

**Land use and land cover change based on historical space-time model**
Qiong Sun, Chi Zhang, Min Liu, Yongjing Zhang*
Tourism Institute of Beijing Union University, Beijing, 100101, China
**Abstract**
Land use and cover change is a leading edge topic in the current research field of global environmental
changes and case study of typical areas is an important approach understanding global environmental
changes. Taking Qiantang River (Zhejiang, China) as an example, this study explores automatic
classification of land use using remote sensing technology and analyzes historical space-time change by
remote   sensing monitoring, which provide new methods for optimizing land use structure and realize
the optimal allocation of land resources as well as intensive utilization. It is of great importance to the
sustainable development of Qiantang River basin and the whole Zhejiang province. This study combines
spectral angle mapping (SAM) with multi-source information and creates a convenient and efficient high
precision land use computer automatic classification method which meets the application requirements
and is suitable for complex landform of the studied area. This work analyzes the histological space-time
characteristic of land use and cover change in 2001, 2007 and 2014, providing a strong information
support and new research method for optimizing Qiantang River land use structure and achieving
optimal allocation of land resources and scientific management.

Key words: Qiantang River; land use; land cover; space-time model; remote sensing technology
**1   Introduction**
Land use refers to all human land development and use activities on purpose, such as, agricultural land,
forestry land, industrial land, land for transportation and residential land (Olang et al., 2014; Ochoa et al.,
2016; Muñoz-Rojas et al., 2015). Land use is closely related to land cover, in which, the former occurs on
the surface of the earth and the latter is the product of various surface processes including land use
(Verburg et al., 2014). Land use and land cover have particular time and space attribute, and its form and
feature change in a variety of space and time scales, which generate a series of ecological
environmental effects (de Mûelenaere et al., 2014).
With the development of science and technology nowadays, human is able to acquire earth observation
remote sensing data and the understanding of earth surface evolves to a new stage, which provide a
more powerful and convenient way of acquiring land use and land cover information, and land use and
land cover mapping has gained the most widely application in the satellite earth observation field
(Gessesse et al., 2015). The methods for studying remote sensing data mainly include static remote
sensing image analysis method and dynamic remote sensing image analysis method. Static remote
sensing image analysis refers to analyzing land cover distribution and changes in different periods
through processing remote sensing data in some fixed time phase based on field investigation or



historical data and then dividing them into different categories. Dynamic remote sensing image analysis method refers to analyzing land cover information in different periods by comparing remote sensing data in different time phases. The method is usually used for studying land cover condition in the period when remote sensing data has been existed, because remote sensing data has only existed for decades. Research on land use and land cover is closely associated with the development of mapping and remote sensing technology (Gelaw et al., 2015; Zhang et al., 2000). Remote sensing has large advantage when being applied in researches on land use because it can observe the whole picture of an area simultaneously or observing the same area repeatedly. Remote sensing can observe and monitor rapidly changing system, for instance, land-marine-atmosphere energy exchange, ocean current, atmospheric ozone, etc., as well as changing system in a slow way (Ferreira et al., 2015; Amuti and Luo, 2014). By using spectral angle mapping (SAM) and multi-source information, this study analyzes land use and land cover in Qiantang River based on historical space-time model, aiming to provide a powerful information support for the optimization of land use structure in the Qiantang River in Zhejiang and the reasonable allocation of land resource and a new approach for research analysis.

## 2    Materlals and methods

The Qiantang River basin has complicated landforms, with 70% of mountains and hills, 30% of plain and basin and 10% of rivers and lakes. The Qiantang River follows through mountainous and hilly land in western Zhejiang. Except the northeast side which faces with the East China Sea, the other sides are surrounded by mountains. It is separated by north-east trend mountain chains. The basin is high in the southwest and low in the northeast and covered by many hills and few plains. Height of more than 10 mountains around and inside the basin is between 1500 and 1800 m, and most of watersheds are 1000 - 1400 m high.

The Qiantang River basin is located in middle subtropical zone, near ocean, and has frequent monsoon activities. Winter is sunny and cold; spring is dominated by rainy days and rainy season comes in March and is over in June; July and August is a period with high temperature and drought, typhoon and rainstorm appear frequently; autumn usually has fresh air and invigorating climate (Xu et al., 2014). The annual average temperature of the basin is 16.1 ~ 17.7 °C and the annual rainfall capacity is 1200 ~ 2200 mm (Xia et al., 2014).

Affected by superior hydrothermal conditions and complex topography, plants in the Qiantang River basin are rich in species and types. Zonal vegetation in the basin is mid-subtropical evergreen broad-leaved forest. A majority of native forest vegetations have been destroyed due to human activity and interference for thousands of years, and some secondary natural evergreen broad-leaved forests survive only in local district where has inconvenient transportation and steep slope (Xia et al., 2016).

The Qiantang River, 605 km long, originated from Xiuning Country of Anhui province (China) crosses Anhui, Zhejiang, Jiangxi and Fujian (China). The Qiantang River abundant in water (average runoff: 43.458 billion m³) has various functions in electricity generation, flood control, drink, cultivation, irrigation, transportation, visit, etc. Main streams of the Qiantang River basin, 583 km long, are made up of Qiantang River, Fuchun River, Xin'an River, Lan River, Heng River, Changshan port and Majin rivulet.





Main streams above Fuchun power station are mountain-rivers with steep slope and hurry flow, and tidal
river reaches are below Fuchun River, with large tidal range in the estuary, which belongs to strong tidal
estuary (Su et al., 2011).

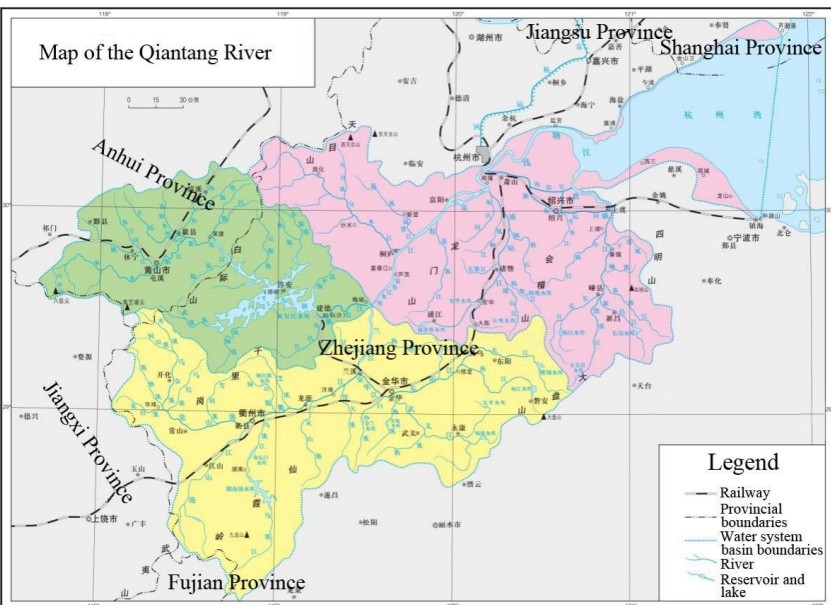


Figure 1 Map of the Qiantang River

**2.2 Land use and cover automatic classification method based on spectral angle mapping**
**and multi- source information**
Multi-source information
(1) Topographic data
Digital line graph (DLG) (1:50000) is used in this study as a topographic data source. Relying on
ArcGIS analysis function (Xiao et al., 2012), triangular irregular network (TIN) can be generated from
DLG and then transformed into digital elevation model (DEM). Then, the gradient and slope aspect
information are extracted.

(2) Normalization indexes
Normalized differential vegetation index (NDVI), the optimal indicative factor for plant growth condition
and spatial distribution density, is lineally associated with plant distribution cover degree, which is the
most widely used vegetation index. Water reflection is weakened gradually from visible light to
middle-infrared band, and water has the strongest absorption in near-infrared and middle-infrared band
and almost reflects nothing. Zhao et al (Zhao and Chen, 2005) compared difference value using





TM/ETM + the 5th and 6th band and also established normalized difference bare index (NDBaI) in the
study of TM/ETM + bare soil image extraction.

Classification methods
(1) Basic principle of **spectral angle mapping**
Spectral angle mapping confirms the similarity between a test spectrum and a reference spectrum by
calculating their angle (Li et al., 2014). Reference spectrum can be the pixel spectrum extracted from
laboratory or field or images. This method assumes that image data have been cut to "apparent
reflection", that is to say, all dark radiation and path radiation deviations have been eliminated. SAM
confirms the similarity between test spectrum $t_i$ and reference spectrum $r_i$ through the following formula:

$$\alpha = \cos^{-1}\left[ \frac{\sum_{i=1}^{n_j} t_i r_i}{\left(\sum_{i=1}^{n_b} t_i^2\right)^{\frac{1}{2}} \left(\sum_{i=1}^{n_b} r_i^2\right)^{\frac{1}{2}}} \right] \qquad (1)$$


Herein, n stands for the number of wavebands. The formula seems like solving the angle between two
vectors. Spectral reflectance ratio of ground object can be regarded as a vector. If total illumination
increases or decreases, the length of the vector will increase or decrease accordingly, but the angle
direction remains unchanged.

(2) Classification process
This study is designed to acquire corresponding topographic data from digital line graph and extract
various normalized index information from TM image, and then, perform SAM classification and
precision evaluation by recombining various multi- source information on TM image. Mixed division of
paddy field, dry field and woodland, i.e., whether the consistent vector directions of paddy field, dry yield
and woodland in six-dimensional space of original spectral information will induce the low prevision of
image classification or not is analyzed according to precision evaluation results in test area and
verification results in verification area, and finally, conclusions are reached.

(3) Data standardization and training sample selection
Before SAM classification, normalized index information is standardized between 0 and 255 by earth
resource data analysis system (ERDAS) modeling. Samples are selected for training after the optimal
waveband combinations are chosen. Pixel samples (300 dpi) in each area are selected to ensure large
differences between spectral vector angles.

**2.3 Brief introduction of land use and cover analysis method**





(1) Single dynamic degree of land use and cover
Single dynamic degree of land use and cover refers to the number of some kind of land use and cover
change in a certain time in a study area (Sanjuán et al., 2016), and its expression formula is:
$$K = \frac{U_b - U_a}{U_a} \times \frac{1}{T} \times 100\%$$
(2)

Where k stands for the dynamic degree of some kind of land use and cover in a certain study time; $U_a$
stands for the number of some kind of land use and cover at the beginning of the study; $U_b$ stands for the
number of some kind of land use and cover types in the end of the study; T stands for the length of
research period. K is considered as annual changing rate of some kind of land use and cover type in the
study area when T is set as year.
(2) Dynamic degree of comprehensive land use and cover
Dynamic degree of comprehensive land use and cover refers to the number of land use change in a
certain time in a study area, and its expression formula is:
$$LC = \left[ \frac{\sum_{i=1}^{n} \Delta LU_{i-j}}{2\sum_{i=1}^{n} LU_i} \right] \times \frac{1}{T} \times 100\%$$
(3)

Herein, $LU_i$ stands for the area of ith land use and cover type at the beginning of monitoring; $LU_{i-j}$ stands
for the absolute value of i land use and cover type transforming into non-i land use and cover type in the
monitoring time; T is study phase. LC value is considered to be annual changing rate of land use and
cover in the study area when T is set as year.
(3) Comprehensive index of land use and cover
Comprehensive index of land use and cover in a study area can be expressed as:
$$L_j = 100 \times \sum_{i=1}^{n} A_i \times C_i$$
(4)

Where $L_j$ stands for comprehensive index of land use and cover in a study area; $A_i$ stands for grading
index of level i land use and cover in the area; $C_i$ stands for area percentage of level i land use and cover
grading in the area; n stands for the number of land use and cover grading.
(4) Analysis of change degree of land use and cover
The change of land use and cover in a certain range is the result of changes of various types of land use
and cover types, and land use and cover as well as its variation and change rate can quantificationally
reveal the overall level and change trend of land use and cover in the range (Yu et al., 2014; Belay et al.,
2015). Variation and change rate of land use and cover can be expressed as:
$$\Delta L_{b-a} = L_b - L_a = 100 \times \left( \sum_{i=1}^{n} A_i \times C_{ib} - \sum_{i=1}^{n} A_i \times C_{ia} \right)$$
(5)

$$R = \frac{\sum_{i=1}^{n} (A_i \times C_{ib}) - \sum_{i=1}^{n} (A_i \times C_{ia})}{\sum_{i=1}^{n} (A_i \times C_{ia})}$$
(6)

Herein, $L_a$ stands for regional land use and cover comprehensive index at time a; $L_b$ stands for regional





land use and cover comprehensive index at time b; $A_i$ stands for level i land use and cover grading index;
$C_{ia}$ stands for area percentage of level i land use and cover at time a in an area; $C_{ib}$ stands for area
percentage of level i land use and cover at time b in an area; $L_{b-a}$ stands for variable quantity of land use
and cover; R stands for change rate of land use and cover.
(5) Information entropy of land use and cover structure
"Entropy", a concept of thermodynamics, is considered as a random variable without restriction in
information theory (Sato and Suganuma, 2013). The size of entropy can be used to describe average
uncertainty degree in probability system and analyze complex land use and cover structure with the help
of the concept of entropy in a thorough and quantitative way. Information entropy (H) is defined as
follows based on Shannon entropy formula:
$$H = -\sum_{i=1}^{n} P_i \times InP_i \qquad (7)$$
Where information entropy H is used to describe the diversity of land use and cover; $P_i$ stands for the
proportion of land type i. The diversity index is considered as 0 when the area has not been developed,
i.e., $H_{min}=0$; various land types have been stable and meet entropy maximization conditions and the
diversity index is maximum when the area has been fully developed, i.e., $H_{max}=InN$ (n stands for land use
and cover types).
(6) Degree of balance and dominance
Information entropy of land use structure is calculated according to actual number of functions, and the
value is usually comparable. Therefore, it is quite necessary to introduce the concept of degree of
balance (Zhu et al., 2008). Based on information entropy formula, degree of balance is expressed as:
$$J = \frac{H}{H_m} = -\left[\sum_{i=1}^{n} P_i \times InP_i\right] and InN$$
$$I = 1 - J \qquad (8)$$
Where J stands for degree of balance, E $\in$ [0, 1], urban land use and cover is in an uneven state when E
is equal to zero, and land use and cover types reach an ideal and balanced situation when E is equal to 1.
I stand for degree of dominance, the larger degree of dominance tends to show larger mean value of
land use and cover and more balanced land distribution. Hence, compared with information entropy, the
index is more intuitive and comparable (Garedew et al., 2009).

**3. Results**
**3.1 Mathematical model analysis methods for land use and cover**

Single dynamics of land use and cover types in the Qiantang River basin from 2001 to 2007, 2007 to
2014 and 2001 to 2014 is shown in table 1. As a whole, the number of paddy field and dry land is
reduced, while forest land, water area and building land increase from 2001 to 2014, in which, building
land changes the fastest while forest land changes the slowest.



It can be seen from dynamic degree index of comprehensive land use and cover in table 1 that the
number of dry land changes greatest from 2001 to 2014, followed by building land and paddy field, and
forest land changes is the minimum. As a whole, the rate and number of building land change fastest and
frequently from 2001 to 2014; dry land changes slowly but the number of dry land changes greatest; the
changing rate and number of paddy field occupy the third place among all types of land. Therefore, dry
land and paddy field belong to sensitive land types in the Qiantang River basin. Water area changes fast,
but the number changes little, which is consistent with season-related change in water area. The change
speed of forest land change is the slowest and its number change is also the most unobvious, which is
consistent with the fact that the total area of forest is the largest and the roll-in and roll-out changes are
mild. Details are shown in figure 2.
Table 1 Dynamic degree of land use and cover types in the Qiantang River basin

| Year | Types (km²) | Dynamic degree of single land use and cover (%) | Dynamic degree of comprehensive land use and cover (%) |
|---|---|---|---|
|  | Paddy field | -1.42 | 8.01 |
|  | Dry land | -2.31 | 11.13 |
| 2001-2007 | Forest land | 0.18 | 0.39 |
|  | Water area | 3.05 | 1.87 |
|  | Building land | 4.71 | 7.52 |
|  | Paddy field | -1.23 | 8.57 |
|  | Dry land | -2.96 | 12.06 |
| 2007-2014 | Forest land | 0.29 | 0.49 |
|  | Water area | 4.53 | 1.61 |
|  | Building land | 6.12 | 5.96 |
|  | Paddy field | -1.29 | 4.23 |
|  | Dry land | -2.56 | 7.98 |
| 2001-2014 | Forest land | 0.25 | 0.23 |
|  | Water area | 3.72 | 0.89 |
|  | Building land | 5.93 | 4.82 |


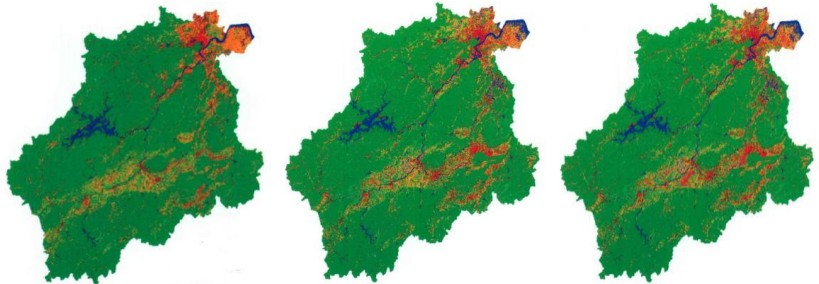

Figure 2 Spectral angle mapping based automatic classification of land use and cover of the Qiantang



River basin in year 2001, 2007 and 2014 (orange: paddy field; light green: dry field; dark green: forest
land; dark blue: water area; red: construction land)

**3.2 Analysis of land use and cover degree**
As shown in table 2, the Qiantang River basin had relatively higher land use and cover, and land use and
cover comprehensive index increased to 233.3582 in 2014 from 232.8926 in 2001. It indicated that land
use and cover comprehensive index increased gradually from 2001 to 2014, which suggested that land
use and cover in the basin were deepening from 2001 to 2014.
It can also be known from table 2 that land use and cover in the Qiantang River basin developed and
deepened continuously in two stages (2001-2007 and 2007-2014), and land use and cover change rate
was larger from 2001 to 2007 than from 2007 to 2014, which showed that the Qiantang River basin
developed rapidly from 2001 to 2007.

Table 2 Land use and cover comprehensive index, variation and change rate in the Qiantang River basin

| Comprehensive index of land use and cover | | | Land use and cover variation quantity | | | Land use and cover degree change ratio (%) | | |
|---|---|---|---|---|---|---|---|---|
| | 2001 | 2007 | 2014 | 2001-2007 | 2007-2014 | 2001-2014 | 2001-2007 | 2007-2014 | 2001-2014 |
| **The basin** | 232.8926 | 233.0125 | 233.3582 | 0.1562 | 0.1249 | 0.2811 | 0.0602 | 0.0816 | 0.1418 |


**3.3  Information entropy and balance degree of land use and cover structure**
Calculation results of land use and cover structural information entropy in 2001, 2007 and 2014 are
displayed in table 3.
It could be seen from table 3 that land use and cover information entropy in the Qiantang River basin
touched the bottom in 2001, suggesting that land use and cover system had a higher degree of order
and stronger constitutive property at that time. However, it reached the highest in 2014, which indicated
that land use system increased in degree of disorder and had the minimum degree of order and poor
constitutive property. Monotonic increasing land use and cover information entropy elaborated that land
use and cover system in the area developed to a relatively disordered state, and land use and cover
structure became more complicated.
As shown in table 3, degree of balance of land use and cover structure in the Qiantang River basin from
2001 to 2014 increases continuously and degree of dominance is reduced, which indicated that land use
and cover structure in the Qiantang River basin is more complicated, the degree of balance is higher and
lands are more evenly distributed as economy develops rapidly. To date, land use and cover structure in
the Qiantang River basin becomes more and more harmonious.







Table 3 land use and cover structure, information entropy and degree of balance and dominance in the Qiantang
River basin in 2001, 2007 and 2014

|  | 2001 | 2007 | 2014 |
|---|---|---|---|
| **Paddy field%** | 12.96 | 11.56 | 10.43 |
| **Dry land%** | 11.86 | 10.12 | 9.63 |
| **Forest land%** | 67.93 | 68.45 | 69.15 |
| **Water area%** | 3.85 | 4.06 | 4.37 |
| **Building land** | 4.09 | 5.02 | 5.98 |
| **Number of functions** | 5 | 5 | 5 |
| **Information entropy** | 1.0203 | 1.0296 | 1.0312 |
| **Degree of balance** | 0.6351 | 0.6332 | 0.6362 |
| **Degree of dominance** | 0.3639 | 0.3623 | 0.3601 |


**4 Discussion**
Currently, research on changes of land use in China concentrates on area with active human activities
and natural motivation, especially developed areas such as Beijing, Yangtze River delta and Shenzhen
and fragile environmental area under the effects of population increase area, economical development
and resource consumption such as northeast China region and Yulin region in transitional zone between
arid and semiarid regions. Zhejiang Qiantang River researched in this study belongs to the first category.
Differing from those hot research area such as Guangzhou and Shanghai (Fan et al., 2007; Yin et al.,
2011), Qiantang River is seldom researched. The Qiantang River basin locating in the west of Zhejiang
province is one of Zhejiang top eight river systems and also the largest river in Zhejiang province.
Moreover, the basin has rich agricultural resources and a long development history. It is always the
important area for comprehensive development of agriculture, forest, grazing, subsidiary business and
fishing and breeds Zhejiang civilization. Changes of land use and cover in the basin are obvious in the
past decades. Research achievements of this study can guide the transformation of local land type and
help people to utilize land resource better on the premise of natural scenery protection.
Research methods for land use and cover include remote sensing data method, model research method
and field observation method (Iqbal et al., 2014; Trabaquini et al., 2014). This study made an automatic
classification of land use and covers in the Qiantang River and made a time-space analysis on land use
and cover of the Qiantang River from 2001 to 2014. Considering the complex terrain, intensive land use
and frequent changes of land use, we found a simple, efficient and high-precise automatic classification
method based on multi-source data in combination with SAM. Based on the maps for classification of
land use and cover of the Qiantang River in 2001, 2007 and 2014, we made a mathematical model
analysis of land use and cover in the Qiantang River and figured out the rules of land use and cover in
the Qiantang River. The automatic classification method integrating multi-source data and SAM is
applicable to research concerning areas with complex terrain and is expected to provide an orientation
for similar researches.

**5  Conclusion**



The level of land use and cover in the Qiantang River basin is high and being deepened. Land use and
cover information entropy in the Qiantang River basin touches the bottom in 2001, suggesting that land
use and cover system has a higher degree of order and stronger constitutive property. However, it
reaches the highest in 2014, which indicates that land use system increases in degree of disorder and
has the minimum degree of order and poor constitutive property. Degrees of balance of land use and
cover structure in the Qiantang River basin from 2001 to 2014 increases continuously and degree of
dominance is reduced. To date, land use and cover structure in the Qiantang River basin becomes more
and more harmonious.

**Acknowledgements**
This study was supported by a grant from the Science and Technology Project of Beijing Municipal
Education Commission (to Sun Qiong) (No. KM201511417009)

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
