# Peer review of "Land use and land cover change based on historical space-time model"

_Solid Earth, 2016_

## Referee Comment (RC1) · Anonymous Referee #1 · 1 Jul 2016

1) The objectives are not clear and concise. Please, provide the aim and objectives following i), ii), iii),... 2) Fig. 1 is not referenced in the text. Please, proceed. 3) Fig. 2. This figure is completely wrong. Maps always have to include at least their scale (at least the scale graph), legend and north arrow. 4) The introduction and method sections are correct, but not the results, discussion and conclusion ones. Please, proceed like this if possible: -Results. Authors should give more details and information about the study as well as describe them more thoroughly. -Discussion. This section is too short; more discussion is necessary. -Conclusions. Please, rewrite the conclusions replying the aim and objectives of the study following i), ii), iii)... Conclusions are not a summary of the study.
* * *

---

## Referee Comment (RC2) · Anonymous Referee #2 · 12 Jul 2016

Review of "Land use and land cover change based on historical space-time model" by Sun et al.

This paper presents results from the automatic classification of land use using remote sensing technology with the aim of analyzing spatial and temporal patterns of land use changes in the Qiantang River catchment. The manuscript deals with a potentially interesting topic but several critical points must be addressed before making it suitable for publication. My main concern is related to the fact that clear objectives of this study are not clearly stated. The authors claim that "…case study of typical areas is an important approach understanding global environmental changes" but a case study is not an "approach", it just, if well presented and discussed, could help to improve the knowledge on consequences of environmental changes. This study needs at least one clear objective (e.g., development of a new methodological approach to investigate

land use changes using remote sensing technique) to justify the case study otherwise the contribution to the scientific community can not be considered sufficiently original. Furthermore, the automatic classification method based on multi-source data in combination with SAM is defined highly accurate and precise (L. 270-271) but a critical comparison with a non-automatic land use changes maps has not been carried out to support this consideration.

Specific comments:

The manuscript presents only two figures and both, especially Fig. 2, require a strong editing. Fig. 1 is not referenced along the text, green, pink and yellow portions of the catchment are not described in legend or caption and a north arrow is missing. Fig. 2 that is probably the most important figure of the paper is unreadable and land use classification differences can not be appreciated because of the small size of individual images of the catchment. I suggest to modify this figure from an horizontal to a vertical combined image (with a, b, c labels) to enlarge the extent of single images. Legend, scale bar and north arrow are completely missing.

In the introduction more international references are needed. You could reference some relevant works using static and dynamic approaches and provide also some example of application of these techniques to different fields of investigation.

L. 41-42: this paragraph is written in poor English. Please consider to reformulate it.

L. 48-52: Here the objective\s of your work must be clearly stated highlighting also the novelty of your study.

L. 55: a title of the sub chapter is required (i.e., 2.1 Study area)

L. 56-62: here some basic information on the studied catchment are missing. Which is the area? And the mean slope?

L. 56: "complicated landforms"->"complex morphology"

How long is Qiantang River? 605 km (L.74) or 583 km (L. 77)?

L. 75-77: English must be revised here and the discharge is usually expressed in mˆ3/s

L. 92: which is the DEM resolution?

L. 256: remove "area" after increase

L. 260: "is seldom researched"->"has been seldom studied in the past".

Discussion: This section needs to be strengthened by better discussing your results and comparing your findings with literature.

---

## Author Comment (AC1) · 20 Jul 2016

**Land use and land cover change based on historical space-time model**

Qiong Sun, Chi Zhang, Min Liu, Yongjing Zhang*

Tourism Institute of Beijing Union University, Beijing, 100101, China

**Abstract**

Land use and cover change is a leading edge topic in the current research field of global environmental changes and case study of typical areas is an important approach understanding global environmental changes. Taking the Qiantang River (Zhejiang, China) as an example, this study explores automatic classification of land use using remote sensing technology and analyzes historical space-time change by remote sensing monitoring. This study combines spectral angle mapping (SAM) with multi-source information and creates a convenient and efficient high precision land use computer automatic classification method which meets the application requirements and is suitable for complex landform of the studied area. This work analyzes the histological space-time characteristics of land use and cover change in the Qiantang River basin in 2001, 2007 and 2014, in order to i) accurately understand the change of land use and cover as well as historical space-time evolution trend, provide a realistic basis for the sustainable development of the Qiantang River basin, ii) and provide a strong information support and new research method for optimizing the Qiantang River land use structure and achieving optimal allocation of land resources and scientific management.

[revised manuscript text omitted]

**4 Discussion**

Land use and cover data is the basis of investigation on global environmental change and also the key
factor in the study of earth surface activity progress; the exploration involves fields like biochemical
circle, plant biomass distribution, climate change and atmospheric circulation. Many research results
suggested that, recent land use was associated to a large number of industrial and agricultural activities
inevitably; land use has gained a great and fast change in the past fifty years, especially in the width and
depth. These changes are usually along with the economic increase and population boom as well as the
change of production and living (Fuller et al., 2012).

It has been known to all that, the change of economy and population is testing the bearing capacity of
biological natural environment and biological damage and resource exhaustion will occur if it exceeds
the bearing capacity. Therefore, to relieve biological risks, ensure the normal production and living of
people, and implement the concept of sustainable development, various countries have input a large
number of manpower and material resources for relevant studies; various research achievement is
playing a positive role.

[revised manuscript text omitted]

Fuller D. O., Parenti M., Gad A., and Beier J.: Land cover in Upper Egypt assessed using regional and global land cover products derived from MODIS imagery, Remote Sens. Lett., 3:171-180, 2012.

Gelaw, A. M., Singh, B. R., and Lal, R.: Organic Carbon and Nitrogen Associated with Soil Aggregates
and Particle Sizes Under Different Land Uses in Tigray, Northern Ethiopia, Land Degrad. Developm., 26,
690-700, doi: 10.1002/ldr.2261, 2015.
Gessesse, B., Bewket, W., and Bräuning, A.: Model-based characterization and monitoring of runoff and
soil erosion in response to land use/land cover changes in the Modjo Watershed, Ethiopia, Land Degrad.
Developm., 26, 711-724, doi: 10. 1002/ldr. 2276, 2015.
Garedew, E., Sandewall, M., Söderberg, U., and Campbell, B. M.: Land-use and land-cover dynamics in
the central rift valley of Ethiopia, Environm. Managem., 44, 683- 694, 2009.
Iqbal, M. F. and Khan, I. A.: Spatiotemporal Land Use Land Cover change analysis and erosion risk
mapping of Azad Jammu and Kashmir, Pakistan, Egypt. J. Remote Sens. Space Sci., 17, 209-229,
2014.
Li, H., Lee, W. S., Wang, K., Ehsani, R., and Yang, C. H.: 'Extended spectral angle mapping (ESAM)' for
citrus greening disease detection using airborne hyperspectral imaging, Precis. Agricult., 15, 162-183,
2014.
Muñoz-Rojas, M., Jordán, A., Zavala, L. M., De la Rosa D, Abd-Elmabod, S. K., and Anaya-Romero, M.:
Impact of land use and land cover changes on organic carbon stocks in Mediterranean soils, Land
Degrad. Developm., 26, 168-179, doi: 10.1002/ldr.2194, 2015.
Mu, J., Khan, S., and Gao, Z.: Integrated water assessment model for water budgeting under future
development scenarios in Qiantang River basin of China, Irrigat. Drain., 57, 369–384, 2008.
Ochoa, P. A., Fries, A., Mejía, D., Burneo, J. I., Ruíz-Sinoga, J. D., and Cerdà, A.: Effects of climate, land
cover and topography on soil erosion risk in a semiarid basin of the Andes, Catena, 140, 31-42.
doi:10.1016/j.catena.2016.01.011, 2016.
Sanjuán, Y., Gómez-Villar, A., Nadal-Romero, E., Álvarez-Martínez, J., Arnáez, J., Serrano-Muela, M.
P., Rubiales, J. M., Gon-zález-Sampériz, P., and García-Ruiz, J. M.: Linking land cover changes in the
sub-slpine and montane belts to changes in a torrential river, Land Degrad. Developm., 27, 179-189,
doi:10.1002/ldr.2294, 2016.
Sato, T. and Suganuma, M.: Consideration of expression method of the entropy concept: correlation
between the thermodynamic entropy obtained from the molecule movement animation and the
psychological quantity from language expression, Trans. Jpn. Soc. Kansei Eng., 12, 303-309, 2013.

Shen X., Xu H. L., Han Y. C., and Tao C. J.: Study on Water Quality Control of Qiantang River Watershed, Environm. Sci. Managem., 38, 68-71, 2013.

Su S., Zhi J., Lou L., Huang F., Chen X., and Wu J. P. Spatio-temporal patterns and source apportionment of pollution in Qiantang River (China) using neural-based modeling and multivariate statistical techniques, Phys. Chem. Earth Parts, A/b/c 36(9–11):379-386, 2011.

Trabaquini, K., Formaggio, A. R., and Galvã,o L. S.: Changes in physical properties of soils with land use time in the Brazilian savanna environment, Land Degrad. Developm., 26, 397-408, doi: 10.1002/ldr.2222, 2015.

Verburg, P. H., Schot, P. P., Dijst, M. J., and Veldkamp, A.: Land use change modeling: current practice and research priorities, Geojournal, 61, 309-324, 2014.

Xia, F., Liu, X. M., Xu, J., Wang, Z. G., Huang, J. F., and Brookes, P.: Trends in the daily and extreme temperatures in the Qiantang River basin, China, Int. J. Climatol., 35, 6553-6565, 2014.

Xia, F., Liu X. M., Xu J. M., Yu L. J., and Shi Z.: Precipitation change between 1960 and 2006 in the Qiantang River basin, eastern China, Climate Res., 67, 257-269, 2016.

Xiao, J. F., Wang, X. D., and Yao, Y.: Underground pipe network spatial analysis in large plant with ArcGIS, J. Comput. Appl., 32, 2675-2678, 2012.

Xu, Y. P., Ma, C., Pan, S. L., Zhu, Q., and Ran, Q. H.: Evaluation of a multi-site weather generator in simulating precipitation in the Qiantang River Basin, East China, J. Zhejiang Univer. - Sci A: Appl. Phys. Eng., 15, 219-230, 2014.

Yin, J., Yin, Z. E., Zhong, H. D., and Wu, J. P.: Monitoring urban expansion and land use/land cover changes of Shanghai metropolitan area during the transitional economy (1979–2009) in China, Environm. Monitor. Assessm., 177, 609-621, 2011.

Yu, B., Stott, P., Di, X. Y., and Yu, H. X.: Assessment of land cover changes and their effect on soil organic carbon and soil total nitrogen in Daqing prefecture, China. Land Degrad. Developm., 25, 520-531, doi: 10.1002/ldr.2169, 2014.

Zhang, F., Tiyip, T., Feng, Z. D., Kung, H. T., Johnson, V. C., Ding, J. L., Tashpolat, N., Sawut, M., and Cui, D. W.: Spatio-temporal patterns of land use/cover changes over the past 20 years in the middle reaches of the tarim river, Xinjiang, China. Land Degrad. Developm., 26, 284-299, doi: 10.1002/ldr.2206, 2015.

Zhao, H. and Chen, X.: Use of normalized difference bareness index in quickly mapping bare areas from

TM/ETM+, Geosci. Remote Sens. Sympos., 3, 1666-1668, 2005.

Zhu, W., Wang, D. H., and Zhou, X. G.: The research of optimizing DEM resolution based on information entropy, Remote Sens. Informat., 18, 79-82, 2008.

---

## Author Comment (AC2) · 20 Jul 2016

(1) comments from Referees Interactive comment on "Land use and land cover change based on historical space-time model" by Qiong Sun et al. Anonymous Referee #2

Review of "Land use and land cover change based on historical space-time model" by Sun et al.

This paper presents results from the automatic classification of land use using remote sensing technology with the aim of analyzing spatial and temporal patterns of land use changes in the Qiantang River catchment. The manuscript deals with a potentially interesting topic but several critical points must be addressed before making it suitable for publication. My main concern is related to the fact that clear objectives of this

study are not clearly stated. The authors claim that ". . .case study of typical areas is an important approach understanding global environmental changes" but a case study is not an "approach", it just, if well presented and discussed, could help to improve the knowledge on consequences of environmental changes. This study needs at least one clear objective (e.g., development of a new methodological approach to investigate

land use changes using remote sensing technique) to justify the case study otherwise the contribution to the scientific community can not be considered sufficiently original. Furthermore, the automatic classification method based on multi-source data in combination with SAM is defined highly accurate and precise (L. 270-271) but a critical comparison with a non-automatic land use changes maps has not been carried out to support this consideration. Specific comments: The manuscript presents only two figures and both, especially Fig. 2, require a strong editing. Fig. 1 is not referenced along the text, green, pink and yellow portions of the catchment are not described in legend or caption and a north arrow is missing. Fig. 2 that is probably the most important figure of the paper is unreadable and land use classification differences can not be appreciated because of the small size of individual images of the catchment. I suggest to modify this figure from an horizontal to a vertical combined image (with a, b, c labels) to enlarge the extent of single images. Legend, scale bar and north arrow are completely missing. In the introduction more international references are needed. You could reference some relevant works using static and dynamic approaches and provide also some example of application of these techniques to different fields of investigation. L. 41-42: this paragraph is written in poor English. Please consider to reformulate it. L. 48-52: Here the objective\s of your work must be clearly stated highlighting also the novelty of your study. L. 55: a title of the sub chapter is required (i.e., 2.1 Study area) L. 56-62: here some basic information on the studied catchment are missing. Which is the area? And the mean slope? L. 56: "complicated landforms"->"complex morphology"

How long is Qiantang River? 605 km (L.74) or 583 km (L. 77)? L. 75-77: English must

be revised here and the discharge is usually expressed in mËĘ3/s L. 92: which is the DEM resolution? L. 256: remove "area" after increase L. 260: "is seldom researched"->"has been seldom studied in the past". Discussion: This section needs to be strengthened by better discussing your results and comparing your findings with literature.

(2) author's response The objectives of the paper have been clarified. Besides, multi-source data based automation classification method in combination with SAM was analyzed.

The content has been corrected according to the above suggestions: Figure 2 has been reedited; literature, illustration and arrow pointing at north have been added for figure 1; literatures in the introduction part have been corrected. But we are sorry that we are unable to correct figure 2 according to the suggestions, but we have demonstrated a map for the Qiantang River basin.

L. 41-42: The language of this section has been revised.

L. 48-52: The research objectives and innovation of the study have been emphasized in that section.

L. 55: A subtitle has been added.

L. 56-62: The relevant information has been supplemented.

L. 56: "Complicated landforms" has been replaced by "complex morphology". The length of Qiantang Rover has been revised. The former one refers to the whole length and the latter one refers to the whole length of the main stream. They are different.

L. 75-77: The data listed was about average annual runoff, thus the unit is correct and remained unchanged. The language has been revised.

L.92ïijŽIt has been pointed out in the study.

L. 256: The word "area" has been removed.

L. 260: "is seldom researched" has been replaced by "has been seldom studied in the past".

The content in the discussion part has been improved.

Please also note the supplement to this comment:
http://www.solid-earth-discuss.net/se-2016-70/se-2016-70-AC2-supplement.pdf